# Diagnostic Performance of Circulating miRNAs and Extracellular Vesicles in Acute Ischemic Stroke

**DOI:** 10.3390/ijms23094530

**Published:** 2022-04-20

**Authors:** Ceren Eyileten, Daniel Jakubik, Andleeb Shahzadi, Aleksandra Gasecka, Edwin van der Pol, Salvatore De Rosa, Dominika Siwik, Magdalena Gajewska, Dagmara Mirowska-Guzel, Iwona Kurkowska-Jastrzebska, Anna Czlonkowska, Marek Postula

**Affiliations:** 1Department of Experimental and Clinical Pharmacology, Centre for Preclinical Research and Technology (CePT), Medical University of Warsaw, Banacha 1B Str., 02-097 Warsaw, Poland; djakubik@wum.edu.pl (D.J.); dmirowska@wum.edu.pl (D.M.-G.); mpostula@wum.edu.pl (M.P.); 2Genomics Core Facility, Center of New Technologies (CeNT), University of Warsaw, 02-097 Warsaw, Poland; 3Department of Medical Pharmacology, Cerrahpasa Medical Faculty, Istanbul University-Cerrahpasa, Istanbul 34096, Turkey; andleeb.shahzadi@iuc.edu.tr; 4Department of Cardiology, Medical University of Warsaw, 02-091 Warsaw, Poland; aleksandra.gasecka@wum.edu.pl (A.G.); dominika.siwik@gmail.com (D.S.); gmgajewska@gmail.com (M.G.); 5Biomedical Engineering & Physics, Amsterdam UMC, University of Amsterdam, 1012 Amsterdam, The Netherlands; e.vanderpol@amsterdamumc.nl; 6Vesicle Observation Centre, Laboratory of Experimental Clinical Chemistry, Amsterdam UMC, University of Amsterdam, 1012 Amsterdam, The Netherlands; 7Division of Cardiology, Department of Medical and Surgical Sciences, Magna Graecia University, 88100 Catanzaro, Italy; saderosa@unicz.it; 82nd Department of Neurology, Institute of Psychiatry and Neurology, 02-957 Warsaw, Poland; ikurkowska@ipin.edu.pl (I.K.-J.); czlonkow@ipin.edu.pl (A.C.)

**Keywords:** miR-19a, miR-186, let-7f, platelet extracellular vesicles, leukocyte extracellular vesicles, platelet reactivity, ischemia, stroke severity, prognosis, EVs

## Abstract

Background: Increased inflammation activates blood coagulation system, higher platelet activation plays a key role in the pathophysiology of ischemic stroke (IS). During platelet activation and aggregation process, platelets may cause increased release of several proinflammatory, and prothrombotic mediators, including microRNAs (miRNAs) and extracellular vesicles (EVs). In the current study we aimed to assess circulating miRNAs profile related to platelet function and inflammation and circulating EVs from platelets, leukocytes, and endothelial cells to analyse their diagnostic and predictive utility in patients with acute IS. Methods: The study population consisted of 28 patients with the diagnosis of the acute IS. The control group consisted of 35 age- and gender-matched patients on acetylsalicylic acid (ASA) therapy without history of stroke and/or TIA with established stable coronary artery disease (CAD) and concomitant cardiovascular risk factors. Venous blood samples were collected from the control group and patients with IS on ASA therapy (a) 24 h after onset of acute IS, (b) 7-days following index hospitalization. Flow cytometry was used to determine the concentration of circulating EVs subtypes (from platelets, leukocytes, and endothelial cells) in platelet-depleted plasma and qRT-PCR was used to determine several circulating plasma miRNAs (miR-19a-3p, miR-186-5p and let-7f). Results: Patients with high platelet reactivity (HPR, based on arachidonic acid-induced platelet aggregometry) had significantly elevated platelet-EVs (CD62+) and leukocyte-EVs (CD45+) concentration compared to patients with normal platelet reactivity at the day of 1 acute-stroke (*p* = 0.012, *p* = 0.002, respectively). Diagnostic values of baseline miRNAs and EVs were evaluated with receiver operating characteristic (ROC) curve analysis. The area under the ROC curve for miR-19a-3p was 0.755 (95% CI, 0.63–0.88) *p* = 0.004, for let-7f, it was 0.874 (95% CI, 0.76–0.99) *p* = 0.0001; platelet-EVs was 0.776 (95% CI, 0.65–0.90) *p* = 0.001, whereas for leukocyte-EVs, it was 0.715 (95% CI, 0.57–0.87) *p* = 0.008. ROC curve showed that pooling the miR-19a-3p expressions, platelet-EVs, and leukocyte-EVs concentration yielded a higher AUC than the value of each individual biomarker as AUC was 0.893 (95% CI, 0.79–0.99). Patients with moderate stroke had significantly elevated miR-19a-3p expression levels compared to patients with minor stroke at the first day of IS. (AUC: 0.867, (95% CI, 0.74–0.10) *p* = 0.001). Conclusion: Combining different biomarkers of processes underlying IS pathophysiology might be beneficial for early diagnosis of ischemic events. Thus, we believe that in the future circulating biomarkers might be used in the prehospital phase of IS. In particular, circulating plasma EVs and non-coding RNAs including miRNAs are interesting candidates as bearers of circulating biomarkers due to their high stability in the blood and making them highly relevant biomarkers for IS diagnostics.

## 1. Introduction

For decades, ischemic stroke (IS) has been considered as the second-most common cause of mortality worldwide. IS accounts for 85% of strokes [1,2,3]. Diagnosis and prognosis of stroke requires computed tomography scans and magnetic resonance imaging, which are not cost-effective methods [4]. Therefore, searching for circulating biomarkers became attractive for researchers, as fluid biomarkers are non-invasive, relatively economical, and easy to obtain. Inflammatory mediators, including interleukin-6 (IL-6), matrix metallopeptidase 9 (MMP-9; known to be modulated by the hypoxic stimulus in endothelial cells [5,6]) and C-reactive protein (CRP) were found promising blood-based biomarkers for IS, however their specificity and capability to differentiate the type of strokes remains questionable [7,8].

Increased inflammation activated blood coagulation system, higher platelet activation plays a key role in the pathophysiology of IS [9,10,11]. Activated platelets are common trigger factors of the acute ischemic events [12], and during platelet activation and aggregation process, platelets may cause increased release of several proinflammatory, and prothrombotic mediators, including microRNAs (miRNAs) and extracellular vesicles (EVs) [13,14,15]. MiRNAs are endogenous single stranded small non-coding RNAs (ncRNAs). MiRNAs are post transcriptional regulators that mostly carry out translational repression and/or degrade target messenger RNA (mRNA) expressions [16,17]. MiRNAs are found to be stored in many different types of cells [18,19], including platelets, and they can regulate platelet reactivity by directly and indirectly targeting specific genes and therefore, modifying protein synthesis [20]. Activated platelets secrete miRNAs into the circulation and platelet-derived miRNAs have been described as potential diagnostic and prognostic biomarkers of cardiovascular (CV) and cerebrovascular diseases including IS [21,22,23,24,25].

Cell-derived or EVs, including microparticles and exosomes, are highly exhibit in body fluids such as blood circulation [26,27]. EVs are particles with a phospholipid bilayer that are naturally released by cells and have no functional nucleus. EVs were found to be released from many cell types and were shown to be associated with a variety of conditions including chronic inflammatory diseases as well as acute inflammatory events such as acute IS [14,28]. Platelet-derived EVs are released by platelets upon activation and aggregation process [29]. Activated platelets and platelets EVs form complexes with leukocytes by binding to cell surface P-selectin (CD62+), which stimulates leukocyte activation. Enhanced leukocyte activation increases the release of proinflammatory indicators and leukocyte-derived EVs, and the production of tissue factor, thereby promoting inflammation [30]. Altogether, both EVs and miRNAs seem to modulate the development and progression of IS, however their diagnostic role in IS is not fully understood.

In our previously published bioinformatics analysis, we demonstrated that miR-19a-3p and miR-186 are involved in the regulation of platelet activation and inflammatory response. Moreover, we have presented the potential of let-7f as both a diagnostic and prognostic biomarker in IS based on a detailed literature search [31]. Therefore, in the current study we aimed to assess several circulating miRNAs (miR-19a-3p, miR-186-5p and let-7f) to analyse their diagnostic and predictive utility in patients with acute IS. Moreover, considering emerging evidence regarding the association between elevated concentrations of different subtypes of EVs and acute IS [14], we evaluated the concentrations of EVs from platelets, leukocytes, and endothelial cells in patients with IS.

## 2. Results

### 2.1. Patient Demographics

Participants’ characteristics and patients’ demographics are presented in Table 1. CV risk factors, such as hypertension (64%) and current smoking (39%), were common in the majority of patients.

### 2.2. Platelet Function

Figure 1 shows the platelet reactivity in patients with IS and control patients, including resting platelets (not stimulated with any agonist as negative control) and AA- (ASPI test), ADP- and TRAP-induced platelets. Patients with IS on day-1 had significantly higher response to AA-induced platelet aggregation compared to controls (*p* = 0.001). Patients after 7-days of acute IS had significant reduction of AA-induced platelet aggregation (*p* = 0.008), and no significant difference was found between day-1 and day-7 (*p* = 0.252) (Figure 1). Moreover, no significant difference was found for ADP- and TRAP- induced platelet aggregation tests between control group and patients with IS on day-1 (*p* = 0.769, *p* = 0.226, respectively).

### 2.3. Association between Analysed miRNAs, EVs and Platelet Function

Patients with IS were divided into two subgroups as high and low platelet reactivity, based on baseline AA-induced platelet aggregation. Patients with ≤30 U AA-induced platelet aggregation was defined as normal platelet activation, and >30 U was defined as high platelet reactivity (HPR) [32]. Patients with normal platelet activation had significantly higher miR-186-5p expression levels compared to patients with HPR at day-1 acute-stroke (*p* = 0.034). Seven days after acute stroke, expression levels of miR-186-5p significantly decreased in the same patients with normal platelet reactivity (*p* = 0.036). No significant difference was observed for miR-19a-3p and let-7f (Figure 2).

Patients with HPR had significantly elevated platelet-EVs (CD62+) concentration compared to patients with normal platelet reactivity at the day of 1 acute-stroke (*p* = 0.012). Similarly, patients with HPR had significantly higher leukocyte-EVs (CD45+) concentration compared to patients with normal platelet function at day-1 acute-stroke (*p* = 0.002). No significant difference was observed for the other analysed EVs (Figure 3).

### 2.4. Alteration and Diagnostic Potential of Analysed miRNAs Expressions and EVs Concentrations

Patients with acute IS had significantly higher miR-19a-3p expression compared to control patients (*p* < 0.001). MiR-19a-3p expression levels significantly decreased at 7-days post-stroke, compared to the first day of IS (*p* = 0.028). No significant differences were found for miR-19a-3p expression levels between control patients and patients at day-7 post-stroke (*p* = 0.810) (Figure 4a). Let-7f expression levels were significantly lower in patients with acute IS compared to control (*p* < 0.001), and let-7f expression remained significantly lower at day 7 compared to control (*p* < 0.001) (Figure 4e). Diagnostic values of baseline miRNAs were evaluated with receiver operating characteristic curve analysis. The area under the ROC curve for miR-19a-3p was 0.755 (95% CI, 0.63–0.88) *p* = 0.004, whereas for let-7f, it was 0.874 (95% CI, 0.76–0.99) *p* = 0.0001 (Figure 4b,f).

Patients with acute IS had significantly higher platelet-EVs (CD61+) concentration both at day-1 and day-7 post-stroke compared to control patients (*p* = 0.001, *p* = 0.030, respectively) (Figure 5a). Similarly, leukocyte-EVs (CD45+) was significantly higher in IS patients at day-1 compared to the control group (*p* = 0.005). No significant differences were observed for leukocyte-EVs concentration between control patients and patients at day-7 post-stroke (*p* = 0.104) (Figure 5c). No significant differences were found for endothelial-EVs concentration between control and patients at day-1 and day-7 post-stroke (*p* = 0.419, *p* = 0.737, respectively) (Figure 5e). Baseline concentration of both platelet-EVs and leukocyte-EVs showed diagnostic value for IS in the acute stage through ROC curve analysis. Area under the ROC curve for platelet-EVs was 0.776 (95% CI, 0.65–0.90) *p* = 0.001, whereas for leukocyte-EVs, it was 0.715 (95% CI, 0.57–0.87) *p* = 0.008 (Figure 5b,d).

ROC curve showed that pooling the miR-19a-3p expressions, platelet-EVs, and leukocyte-EVs concentration yielded a higher AUC than the value of each individual biomarker as AUC was 0.893 (95% CI, 0.79–0.99) (Figure 6).

### 2.5. Baseline miR-19a-3p Expression Predicts Severity of Stroke

Among stroke patients, 15 patients (54%) presented minor, and 13 patients (46%) presented moderate stroke at admission. Patients with moderate stroke had significantly elevated miR-19a-3p expression levels compared with patients with minor stroke at the day of acute IS. AUC in ROC curve analysis was 0.867, (95% CI, 0.74–0.10) *p* = 0.001 (Figure 7). No significant difference was observed for miR-186-5p, Let-7f, platelet-EVs, leukocyte-EVs, and endothelial-EVs.

## 3. Discussion

Our results demonstrate that increased platelet activation in acute IS is accompanied by the elevation of both platelets- and leukocytes-derived EVs in the first 24 h after ischemic episode, which decrease to baseline after 7 days. The data also show the time-dependent dynamics of AA-induced platelet reactivity and both types of EVs concentrations during the first week following IS. Apart from increase in EVs concentrations, our results also demonstrate that miR-19a-3p is overexpressed and let-7f is downregulated 24 h after stroke in comparison to controls, but not at day 7. Finally, the combination of EVs (CD45+ and CD61+), miR-19a-3p and let-7f may have a high diagnostic value for acute IS. However, only differences in the expression of miR-19a-3p may help to distinguish severity of stroke according to NIHSS score at day 1 after ischemic episode.

Previously it was demonstrated that EVs derived from endothelial cells EVs (including CD146+) are the most increased in acute IS when compared to high-CV risk controls. Also, platelet- (CD61+) and leukocyte-derived (CD45+) EVs were elevated in the blood samples of acute IS patients [33]. Some other studies confirmed that levels of EVs of different origin are increased in acute IS [34,35,36]. Moreover, in the most recent study Huo et al. found that high levels of both endothelial and leukocyte EVs after acute IS were associated with worse CV outcome in long term follow-up [37]. In the current study only platelets- and leukocytes-derived EVs were higher in acute IS than controls. Similarly, previous research showed higher concentration of leukocytes derived EVs in the acute phase of IS [38]. It was hypothesised that early levels of leukocyte EVs may be enhanced by leukocyte activation. Moreover, CD45+ EVs may play a role as proinflammatory mediators and thus contribute to vascular inflammation [30] and carotid plaque instability [39]. Our study, like others, found a higher concentration of platelet-EVs in the early phase of IS [33,34,35,40,41,42], which confirms the diagnostic potential of platelet EVs. Moreover, we found a higher concentration of both platelet and leukocyte EVs in patients with HPR 24 h after ischemic episode, but not in day 7 which confirms that platelet EVs may be markers of successful antiplatelet and thrombolytic treatment following an acute stroke [33,34,35,40]. In our study, platelet EV concentrations after IS paralleled platelet reactivity changes, as measured with the AA-induced aggregation test, both at 24 h and seven days after ischemic episode, thus further enforcing our findings. Finally, we found that both CD61+ and CD45+ EVs might have diagnostic potential in early phases of IS.

Even though in one of the earliest studies miR-19a was described as the most abundant miRNAs in blood platelets [43,44], we could not confirm its relationship with platelet activation in acute phase of IS. However, in our previously published bioinformatic analysis we described that miR-19a-3p might play a role in several biological processes involved in pathogenesis of IS, including inflammatory response, blood coagulation, and platelet activation [31]. To support our findings, predictive role of miR-19a in future cerebral ischemic events occurrence was observed in patients with asymptomatic carotid artery stenosis (CAS), and miR-19a-3p expression was markedly higher than that in healthy subjects. Moreover, miR-19a-3p was a strong predictor of the severity of CAS [45]. Based on in vitro and in vivo analysis it was shown that miR-19a-3p directly targets and inhibits expression of *IGFBP3*, the gene encoding one of the insulin-like growth factors, and thus induce inflammation and apoptosis processes related to ischemia/reperfusion brain injury. Using an animal model, miR-19a-3p inhibitor reduced ischemic brain injury size and improved neurological status compared with the non-treated group, which is in line with our results that showed a correlation between miR-19a-3p and severity of stroke based on NIHSS at admission. It was also observed that reduced brain cell apoptosis was accompanied by lower levels of proinflammatory cytokines including TNF-α, IL-1β, and IL-6 [46]. Another mechanism mediating ischemic brain injury that might be a direct target of miR-19a-3p is *ADIPOR2* gene encoding adiponectin receptor 2. Previously the relation between ischemic stroke and ADIPOR2 expression has been established, which might result in impaired glycolysis enzymes’ expression, glucose uptake and lactate production, and neuronal apoptosis [47]. Another gene directly regulated by miR-19a-3p is *AMPK1*, an encoding protein participating in AMPK/GSK-3β/HO-1 pathway involved in the cellular redox and energy balance [48,49]. In vitro study showed that the activity of AMPK/GSK-3β/HO-1 pathway was decreased in damaged cells and AMPK expression was inversely correlated with miR-19a-3p. Interestingly, glycine treatment downregulated miR-19a-3p expression and increased AMPK/GSK-3β/HO-1 pathway activity improving cellular metabolism. It is worth noting, that in patients with ischemic stroke, the levels of miR-19a-3p in plasma were up-regulated, indicating its clinical potential [49].

In our study we also found a lower expression of let-7f 24-h and 7-days following IS in comparison to controls. Our data support and extend previous findings on let-7f expression changes after ischemic events and may serve as a diagnostic biomarker of IS. The lower expression of let-7 in plasma may be due to several reasons, including the change in miRNAs in specific cell types and the change in cell population after ischemic injury because of immune mediators infiltration, apoptosis, oxidative stress and fibroblast proliferation [50,51]. Previously, similarly to our observation, it was found that acute ischemia leads to rapid downregulation of let-7f expression in myocardial infarction (MI). It was followed by subsequent Tgfbr3 signaling and p38-induced cardiomyocyte apoptosis. Thus, it was identified the let-7/Tgfbr3/p38 axis as an critical indicator of cardiomyocyte apoptosis after MI [52]. Besides, let-7f expressions were downregulated in patients with CAD when compared to healthy controls [53,54]. Higher expression of let-7f, were associated with reduced accumulating major adverse cardiac and cerebrovascular events after cardiac surgery due to CAD [54]. Recent studies showed that upregulation of let-7f expressions in the ischemic myocardium could ameliorate cardiac repair by triggering pro-angiogenic factors [55]. Its protective role may result from promoting proliferation and angiogenesis of endothelial cells through inhibiting the antiangiogenic transforming growth factor-β/ALK5 pathway [56]. Also, in acute IS using microarray assays expression changes of the let-7 family were described [57,58,59]. For instance, let-7f expression was found lower in 3 subtypes of stroke patients (large artery, small artery, and cardioembolic) compared to the normal controls [60]. Moreover, the expression level of let-7f was significantly lower in massive cerebral infarction (MCI) without hemorrhagic intracerebral haemorrhage patients than in that of the healthy controls and the patients with IS without MCI or intracerebral haemorrhage [61]. However, in our study we could not determine differences between groups based on stroke severity. It was also shown that beside inhibiting inflammatory response by targeting IL-6 gene expression in patients with ischemic stroke, let-7f may suppress expression of gene encoding *MTHFR* leading to low cellular MTHFR levels and excessive production of reactive oxygen species [61,62]. Taken together, in our study, patients with acute IS had significantly higher platelet-EVs concentration both at day-1 and day-7 compared to control patients. Similarly, leukocyte-EVs was significantly higher in acute IS patients at day-1 compared to the control group. These results show the diagnostic potential of both platelet- and leukocyte-EVs in acute IS. Importantly, combining platelet-EVs and leukocyte-EVs together with miR-19a-3p improved the accuracy of diagnostic utility more than the value of each individual biomarker. Therefore, the measurement of multiple biomarkers would be more helpful in the clinical setting, not only to increase the diagnostic capacity of the test, but also provide additional information on the temporal evolution of the stroke event.

The main limitation of our study is the small size of the patient group. Although this number (*n* = 28) is small, this cohort has a representative distribution of clinical characteristics. Secondly, given the hypothesis-generating study design, we limited our analysis to the miRNAs associated with platelet function, based on our previous bioinformatic analysis. We did not perform the miRNAs sequencing in the collected plasma samples, which might enable us to determine novel miRNAs with diagnostic value in IS. Moreover, regarding EVs evaluation, due to low positive events count (<10, which is too little counts for a reliable concentration estimation) CD62+ and CD61, CD62+ events were omitted from the analysis in the control group and they were presented only in the IS patients groups. It is important to note that, CD62+ is a marker for platelet activation, hardly present in plasma of healthy individuals, but abundantly present in plasma of patients with myocardial infarction [63]. Therefore, hardly detected CD62+ platelet-derived EVs in the control group can be due to normal platelet function. Thus, we still suggest that CD62+-exposing EVs are a better marker differentiating patients with IS and controls than CD61+ or CD45+, because the latter are present also in plasma of healthy individuals, albeit at lower concentrations than in plasma of IS patients.

## 4. Methodology

### 4.1. Participants

The study population consisted of 28 patients with the diagnosis of the acute IS based on clinical features according to the World Health Organisation definition and supported by brain imaging (CT or MRI) [64]. Based on the Trial of Org 10172 in Acute Stroke Treatment (TOAST) classification we included only patients classified as having ischemic strokes due to large-vessel atherosclerosis with at least 50% stenosis of carotid artery ipsilateral to the infarct side or cardioembolic stroke in individuals with one major cardiac risk factor for embolism with no evidence of other stroke subtypes [65]. Stroke severity was classified at admission using the National Institutes of Health Stroke Scale (NIHSS), classifying scores of 1–4 as minor and scores of 5–15 as moderate [66,67]. During hospitalization, Duplex Doppler imaging of extracranial arteries, through with 24h Holter ECG monitoring, were performed. Patients with prior stroke and patients who died during hospitalization were excluded from the analysis.

The control group consisted of 35 age- and gender-matched patients on acetylsalicylic acid (ASA) therapy without history of stroke and/or TIA with established stable coronary artery disease (CAD) and concomitant CV factors.

The study protocol and informed consent form, designed in compliance with the Declaration of Helsinki, were approved by the Ethics Committee of the Medical University of Warsaw, Warsaw, Poland (approval number: KB/148/2017, approval date: 4 July 2017). Informed written consent was obtained from all enrolled patients or from their relatives.

### 4.2. Samples Collection and Handling

Venous blood samples were collected from the control group and patients with IS on ASA therapy (a) 24 h after onset of acute IS, (b) 7-days following index hospitalization. Briefly, blood was collected in 10-mL citrated blood collection tubes (S-Monovette, Sarstedt, Hildesheim, Germany) via antecubital vein puncture using a 19-gauge needle, without tourniquet. The first 2 mL were discarded to avoid pre-activation of platelets. Right after blood collection, platelet-depleted plasma was prepared by double centrifugation as previously described [68,69,70]. Supernatant platelet-depleted plasma was transferred into 1.5 mL low-protein binding Eppendorfs (Thermo Fisher Scientific, Waltham, MA, USA) tubes, and stored in −80 °C until analyzed.

At each time point during blood sampling, an additional blood sample was collected to a 2.7-mL hirudin tube (S-Monovette, Sarstedt) to assess platelet function by using multiple electrode aggregometry (MEA, Roche Diagnostics, Basel, Switzerland). Platelet activity analysis was performed only once in the control group and twice in the IS group (24 h and 7-day post-stroke).

### 4.3. Platelet Function Analysis

Platelet reactivity was assessed by MEA with the use of adenosine diphosphate test (ADP, 6.5 µmol/L) and ASPI test (arachidonic acid-AA, 0.5 mmol/L), thrombin receptor-activating peptide-6 (SFLLRN) test (TRAP, 32 µmol/L) in hirudin whole blood, as previously described [69,71]. Unstimulated hirudin whole blood (resting platelets) was used as a negative control.

### 4.4. Extracellular Vesicles Determination

Flow cytometry (A60-Micro, Apogee Flow Systems, Hemel Hempstead, UK) was used to determine the concentration of EV subtypes in platelet-depleted plasma. To prevent swarm detection, samples were diluted 10-fold to 260-fold in phosphate-buffered saline (PBS) before staining. The aim of custom sample dilutions was to achieve a count rate below 5000 events/seconds upon measuring, which prevents swarm detection for our assay and blood plasma samples [72]. We measured the diluted samples 120 s, flow rate was 3.01 μL/min, and trigger threshold was 14 arbitrary units (AU) of the side scatter detector, corresponding to a side scattering cross section of 10 nm^2^. With the concentration of EVs per mL of plasma, we mean particles (i) exceeding the side scatter threshold, (ii) with a diameter > 200 nm as determined with the flow cytometry scatter ratio (Flow-SR) [70], (iii) with a refractive index < 1.42 to omit non-specifically labelled chylomicrons [73], and (iv) positive at the fluorescence detector(s) for the used label(s). Due to the detection limit of our flow cytometer, it is expected that most EVs measured in this study originate from the cell-surface. We labelled EVs released by all platelets and megakaryocytes (CD61+), activated platelets (CD62+), leukocytes (CD45+), and endothelial cells (CD146+). To improve the reproducibility of the flow cytometry analysis, we (i) reported our results according to the standardized framework (MIFlowCyt-EV) [74], (ii) calibrated all detectors, (iii) applied Flow-SR to determine the diameter and refractive index of EVs [70], and (iv) automated data calibration and processing with a custom-built software [69]. Our assay and gating strategy resulted in <10 events positive for CD62+ in 74 out of 82 samples, which are too little counts for a reliable concentration estimate. Therefore, CD62+ and CD61, CD62+ events were omitted from the analysis in the control group (See Appendix A).

### 4.5. RNA Preparation, Detection and Quantification of miRNAs by Quantitative PCR

Plasma RNA was extracted by miRVANA PARIS Kit (Invitrogen, Waltham, MA, USA). Total RNA was reverse transcribed by using the TaqMan miRNA Reverse Transcription kit (Applied Biosystems, Waltham, MA, USA) according to the manufacturer’s protocol. TaqMan miRNA Assay kits (Applied Biosystems) was used to determine the miRNAs expression by quantitative real-time polymerase chain reaction (qRT-PCR) with CFX384 Touch Real-Time PCR Detection System (BioRad Inc., Hercules, CA, USA). During the RNA extraction phase cel-miR-39 was spiked-in as an exogenous control and all qRT-PCRs were normalized to their corresponding cel-miR-39. Reactions were run in triplicate and miRNA expressions were expressed as 2-ΔCT [25,75].

### 4.6. Statistical Analysis

Clinical data and categorical variables are presented as percentages of patients and were compared using χ^2^ tests. The distribution of data was checked with the A Shapiro–Wilk test. Continuous variables were presented as mean and standard deviation or median with interquartile range. Mann-Whitney U test or unpaired-*t* test were used for two independent samples comparison. Comparison between day-1 stroke and day-7 stroke was done by paired-*t* test or Wilcoxon test, depending on the data distribution. Receiver operating characteristic (ROC) curves were used to analyse diagnostic value of miRNAs expressions and in relation to prediction of stroke severity. MiRNAs expression data was log10-transformed for statistical analysis. All tests were two-sided, and *p*-value < 0.05 was considered statistically significant. Multiple comparisons were performed using the Kruskal Wallis or One-way ANOVA with Tukey’s HSD post hoc test, depending on the data distribution. Median and range of log10-transformed miRNA expressions and EVs concentration were presented by box-plots. Statistical analyses were performed by using SPSS version 22.0 (IBM Corporation, Chicago, IL, USA) and GraphPad Prism software version 8.0 (GraphPad Software, San Diego, CA, USA).

Sample size calculation was based on the expected concentration and variance of the pre-selected miRNAs. Thus, we calculated that 22 patients per arm would be required to have a 90% chance of detecting, as significant at the 5% level, a 20% change in the primary outcome measure between the experimental and control groups. To make up for possible methodological issues related to sample storage or other analytical steps, the sample size was arbitrarily increased by 20% to reach 28 subjects per arm. To account for the higher variance expected in the control group, its size was further increased by 25%.

## 5. Conclusions

Our data indicate that combining different biomarkers of processes underlying IS pathophysiology might be beneficial for early diagnosis of ischemic events. Thus, we believe that in the future circulating biomarkers might be used in the prehospital phase of IS. In particular, EVs and ncRNAs including miRNAs are interesting candidates as bearers of circulating biomarkers due to their high stability in the blood and making them highly relevant biomarkers for IS diagnostics. Despite methodological challenges, EVs and miRNAs could provide potential benefits as biomarkers, diagnostic tools, and prediction of IS severity. Further prospective studies should validate sensitivity and specificity and replicate diagnostic performance of these biomarkers in a large population of stroke patients to advance our knowledge and identify new molecular targets for translational research and development of novel specific management and therapeutic strategies.

## Figures and Tables

**Figure 1 ijms-23-04530-f001:**
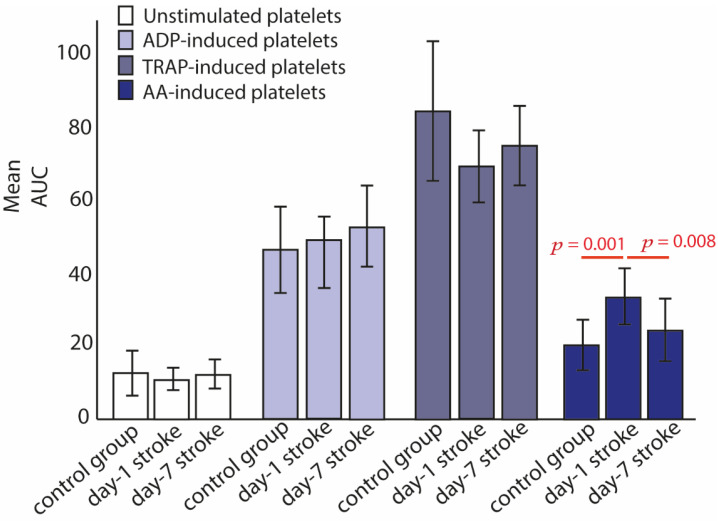
Platelet reactivity in blood in response to arachidonic acid (AA-test), adenosine diphosphate (ADP test), and thrombin receptor-activating peptide-6 (TRAP test). Unstimulated platelets (no agonist) were used as a negative control. *p* value was calculated with paired or unpaired *t* test, appropriately (Control samples *n* = 35, acute stroke day-1 samples *n* = 28 and stroke day-7 samples *n* = 20). Only statistical significant *p* values between the groups are shown (i.e., *p* ≤ 0.05). Abbreviations: AUC, area under the curve.

**Figure 2 ijms-23-04530-f002:**
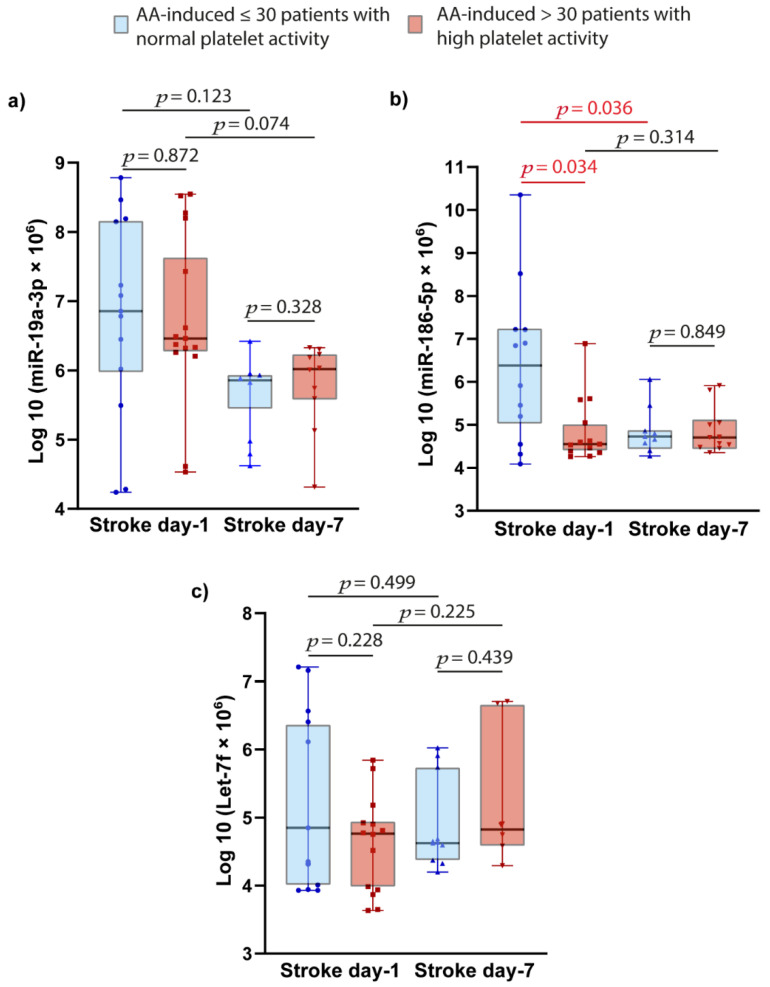
Relation between platelet activation and plasma miRNAs expressions evaluated in plasma. (**a**) miR-19a-3p; (**b**) miR-186-5p; (**c**) Let-7f. *p* value was calculated with Mann-Whitney U or Wilcoxon test, appropriately. Abbreviations: AA, arachidonic acid; Log10, log10 transformation; miR, microRNA; *p*, value.

**Figure 3 ijms-23-04530-f003:**
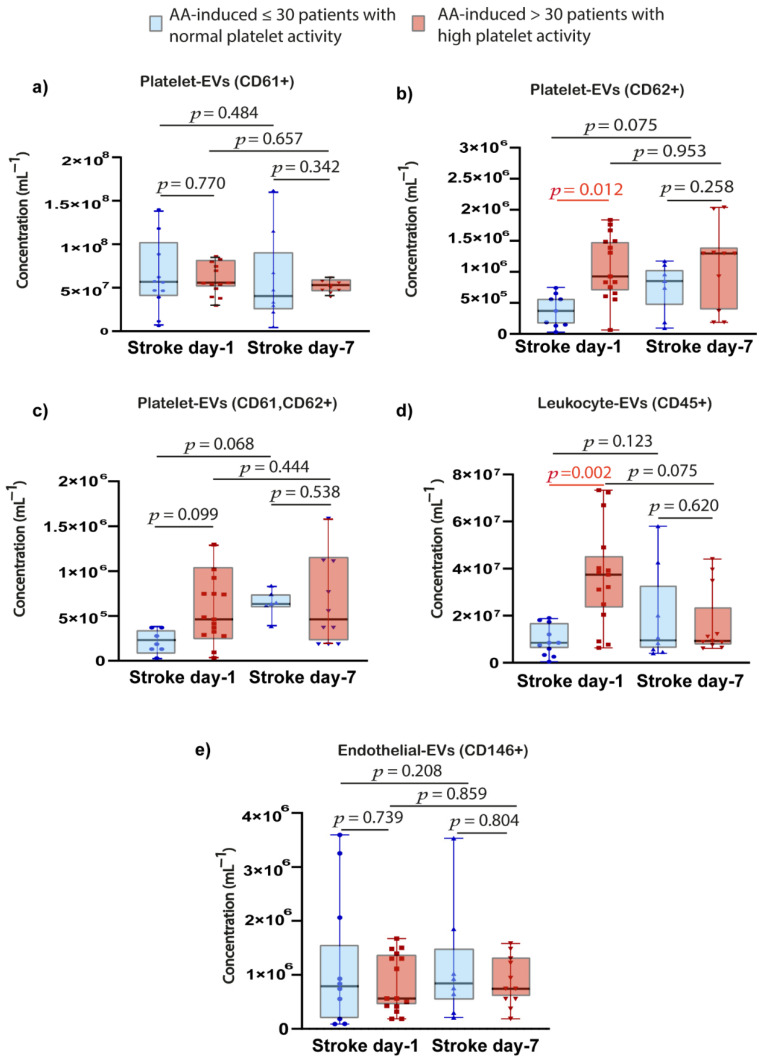
Relation between platelet activation and plasma EVs concentrations. (**a**) Platelet-EVs (CD61+); (**b**) Platelet-EVs (CD62+); (**c**) Platelet-EVs (CD61, CD62+); (**d**) Leukocyte-EVs (CD45+); (**e**) Endothelial-EVs (CD146+). *p* value was calculated with Mann-Whitney U or Wilcoxon test, appropriately. Abbreviations: AA, arachidonic acid; Log10, log10 transformation; miR, microRNA; EVs, extracellular vesicles; *p*, value.

**Figure 4 ijms-23-04530-f004:**
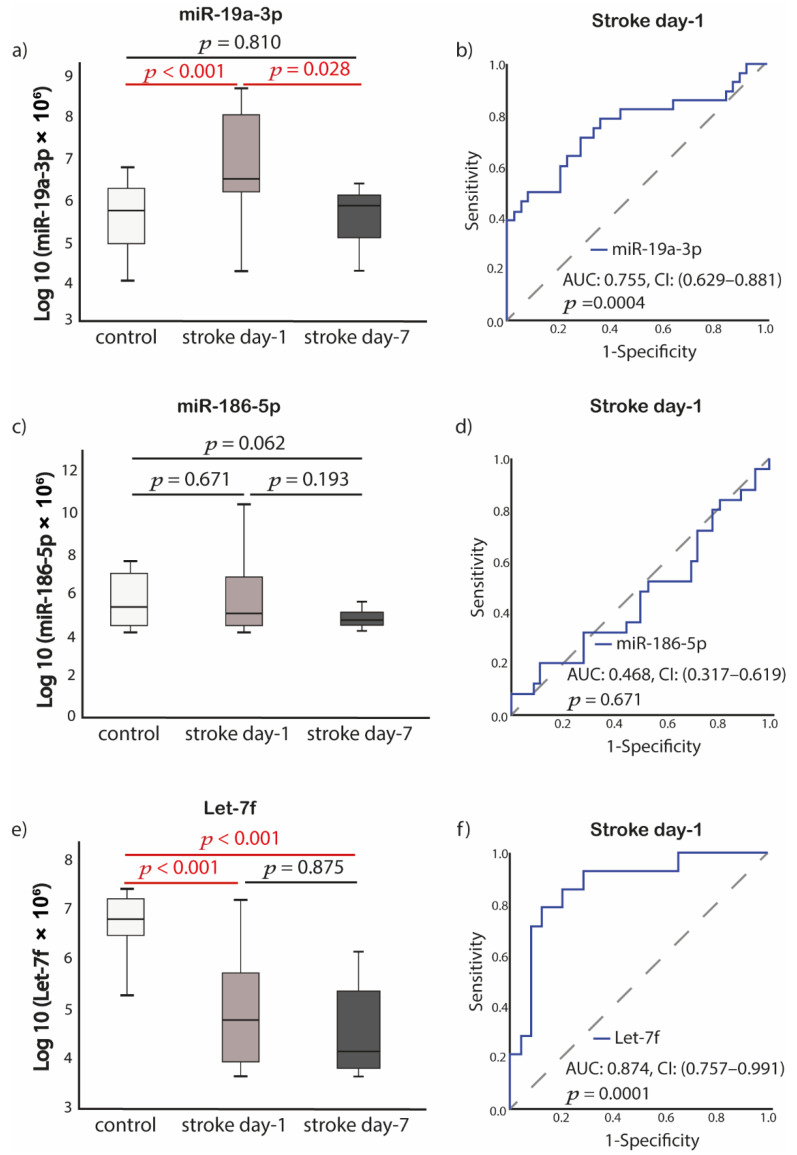
Alterations of miRNAs expression among the groups and diagnostic value of miRNAs determined by ROC curve analysis. (**a**) miR-19a-3p; (**b**) ROC analysis of miR-19a-3p from stroke day-1; (**c**) miR-186-5p; (**d**) ROC analysis of miR-186-5p from stroke day-1; (**e**) Let-7f; (**f**) ROC analysis of Let-7f from stroke day-1. *p* value was calculated with Mann-Whitney U or Wilcoxon test, appropriately (Control samples *n* = 35, acute stroke day-1 samples *n* = 28 and stroke day-7 samples *n* = 20). Abbreviations: Log10, log10 transformation; miR, microRNA; *p*, value; AUC, area under the curve; CI, confidence interval; ROC, Receiver operating characteristic.

**Figure 5 ijms-23-04530-f005:**
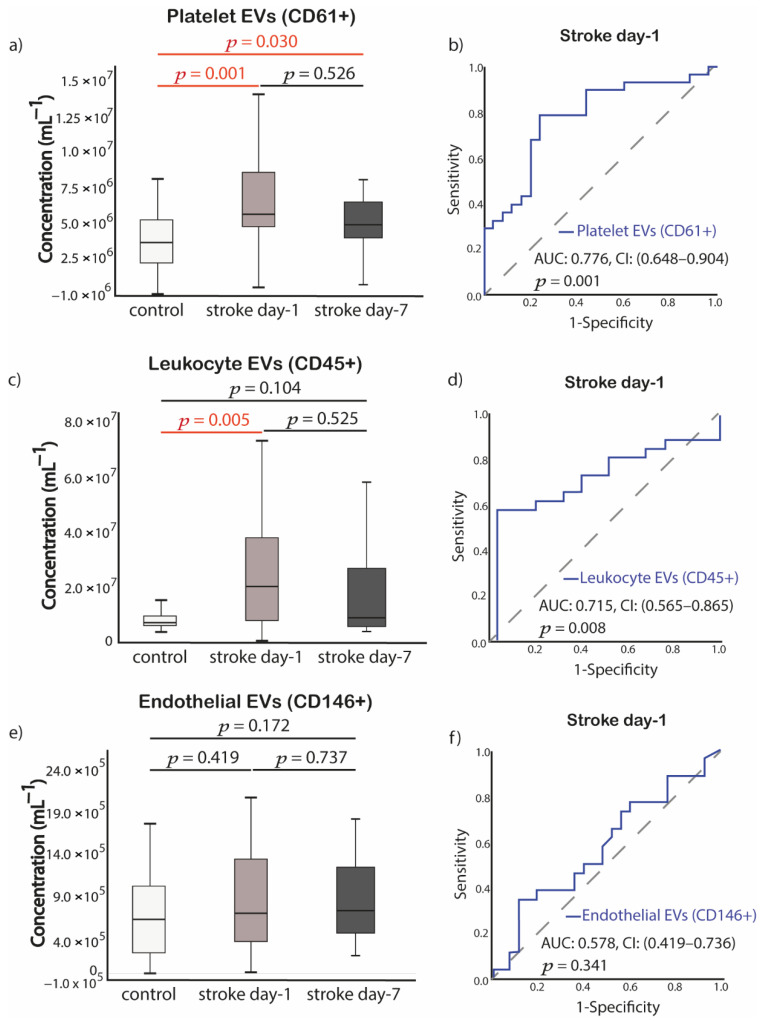
Alterations of EVs concentrations among the groups and diagnostic value of EVs determined by ROC curve analysis. (**a**) Platelet-EVs (CD61+); (**b**) ROC analysis of platelet-EVs (CD61+) from stroke day-1; (**c**) Leukocyte-EVs (CD45+); (**d**) ROC analysis of leukocyte-EVs (CD45+) from stroke day-1; (**e**) Endothelial EVs (CD146+); (**f**) ROC analysis of endothelial EVs (CD146+) from stroke day-1. *p* value was calculated with Mann-Whitney U or Wilcoxon test, appropriately (Control samples *n* = 35, acute stroke day-1 samples *n* = 28 and stroke day-7 samples *n* = 20). Abbreviations: EVs, extracellular vesicles; *p*, value; AUC, area under the curve; CI, confidence interval; ROC, Receiver operating characteristic.

**Figure 6 ijms-23-04530-f006:**
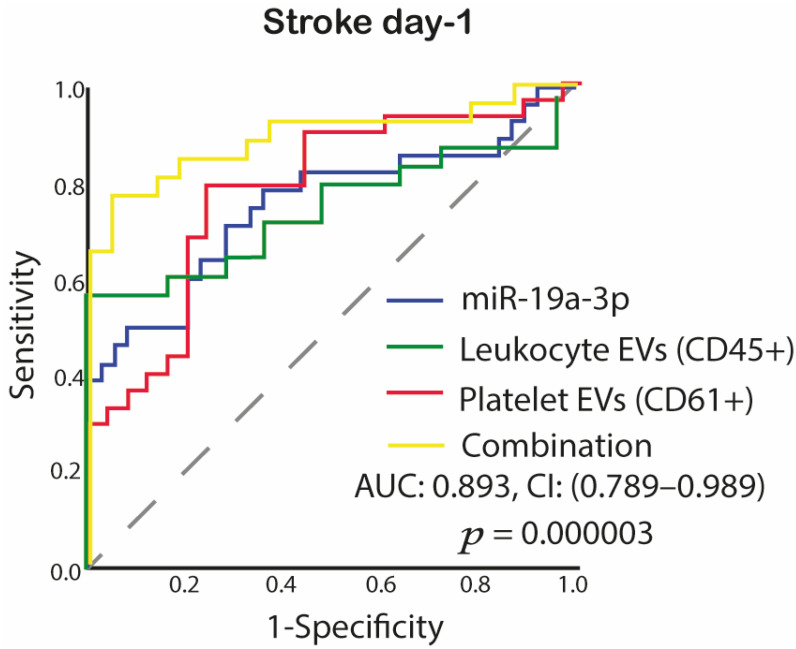
The ROC curve evaluates the diagnostic value of combined miRNA and EVs in the model in acute IS. ROC curves of the three significantly deregulated studied biomarkers (miR-19a-3p, leukocyte-EVs and platelet EVs) showed high diagnostic accuracy. Abbreviations: EVs, extracellular vesicles; miR, microRNA; *p*, value; AUC, area under the curve; CI, confidence interval; ROC, Receiver operating characteristic.

**Figure 7 ijms-23-04530-f007:**
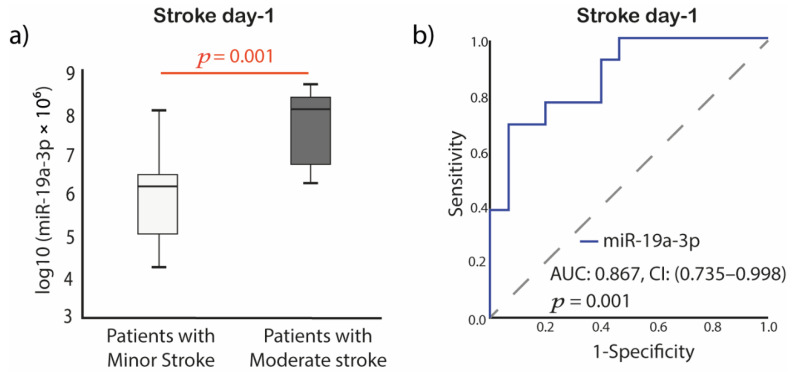
MiRNAs expression difference based on stroke severity at the day of stroke. (**a**) miR-19a-3p box plots; (**b**) miR-19a-3p ROC curves for prediction of stroke severity. The initial stroke severity as measured by the NIHSS at admission. Minor stroke: 1–4 (*n* = 15) and moderate stroke: 5–15 (*n* = 13). Abbreviations: log10, log10 transformation; miR, microRNA; *p*, value; AUC, area under the curve; CI, confidence interval; ROC, Receiver operating characteristic; NIHSS, National Institutes of Health Stroke Scale.

**Table 1 ijms-23-04530-t001:** Participants characteristics.

	**Control (*N* = 35)**	**Acute IS (*N* = 28)**		***p* Value**
**Baseline characteristics**				
Gender (female, *n*, %)	14 (40%)	13 (46.4%)		0.608
Age (years)	65.09 ± 8.01	66.39 ± 15.92		0.681
Body Mass Index kg/m^2^	26.12 ± 2.31	24.5 ± 3.87		0.101
**Comorbidities**				
Hypertension	22 (63%)	18 (64%)		0.907
History of heart failure	4 (11%)	3 (10.7%)		0.369
History of atrial fibrillation	0 (0%)	3 (10.7%)		0.162
History of type 2 diabetes mellitus	7 (20%)	5 (17%)		0.778
Current smoking	8 (23%)	11 (39.3%)		0.933
Prior myocardial infarction	4 (11.4%)	4 (14%)		0.926
Prior transient ischemic attack	0 (0%)	6 (21%)		**0.004**
Prior coronary artery disease	35 (100%)	8 (28%)		**<0.001**
**Laboratory data**				
HCT	41.86 ± 3.49	39.68 ± 4.92		0.064
WBC (×10^9^/L)	7.53 ± 0.44	8.61 ± 3.34		0.168
High-sensitivity C-reactive Protein (mg/dL)	1.10 [0.40–2.90]	2.25 [1.4–5.48]		**0.016**
Fibrinogen (mg/dL)	376.70 ± 126.70	326.25 ± 90.64		0.094
INR	1.05 ± 0.07	1.08 ± 0.31		0.582
Total cholesterol	-	186.00 ± 59.65		-
HDL	-	53.36 ± 17.21		-
LDL	-	102.37 ± 49.27		-
TG	-	118.50 ± 61.42		-
Infarct size	-	2.07 ± 1.58		-
NIHSS at admission (1–4) (5–15)	-	15 (54%) 13 (46%)		-
**Concomitant treatment before the acute IS**				
ASA	35 (100%)	22 (76%)		**0.004**
Diuretics	26 (74%)	8 (28%)		**<0.001**
Statins	32 (91%)	27 (93%)		0.419
ACEi/ARB	26 (74%)	19 (66%)		0.575
Oral anticoagulants	1 (3%)	4 (14%)		0.095
B-blockers	28 (80%)	11 (38%)		**0.001**
**MiRNAs expressions and EVs concentrations**				
	**Control** **(*N* = 35)**	**Stroke day-1** **(*N* = 28)**	**Stroke day-7** **(*N* = 20)**	***p* value ***
miR-19a-3p **	5.77 [5.02–6.34]	6.55 [6.28–8.18]	5.83 [4.98–5.96]	**<0.001**
miR-186-5p **	4.87 [4.45–7.19]	5.12 [4.84–6.88]	4.70 [4.40–5.45]	0.053
Let-7f-5p **	6.82 [6.30–7.21]	4.76 [3.94–5.81]	4.03 [3.76–5.46]	**<0.001**
CD61+	35 × 10^6^ [21 × 10^6^–52 × 10^6^]	56 × 10^6^ [46 × 10^6^–85 × 10^6^]	50 × 10^6^ [39 × 10^6^–67 × 10^6^]	**0.003**
CD62+	-	65 × 10^4^ [14 × 10^4^–13 × 10^5^]	92 × 10^4^ [18 × 10^4^–12 × 10^5^]	**-**
CD61, CD62+	-	32 × 10^4^ [9 × 10^4^–74 × 10^4^]	38 × 10^4^ [18 × 10^4^–76 × 10^4^]	**-**
CD45+	74 × 10^5^ [60 × 10^5^–102 × 10^5^]	22 × 10^6^ [7 × 10^6^–38 × 10^6^]	92 × 10^5^ [64 × 10^5^–348 × 10^5^]	**0.019**
CD146+	63 × 10^4^ [21 × 10^4^–106 × 10^4^]	65 × 10^4^ [28 × 10^4^–141 × 10^4^]	75 × 10^4^ [55 × 10^4^–130 × 10^4^]	0.391

Data are presented as mean ± standard deviation (SD) and median and interquartile range based on the data distribution and percentages. *p* values marked with bold indicate statistically significant differences between the groups < 0.05. * Multiple groups comparison *p* value was calculated based on One-way ANOVA or Kruskal-Wallis test, depending on the data distribution. ** miRNAs expression data was presented as log10 transformation. Abbreviations: HCT, haematocrit; WBC, white blood cell; HDL, high-density lipoprotein; LDL, low-density lipoprotein; TG, triglyceride; NIHSS, National Institutes of Health Stroke Scale; ASA, acetyl salicylic acid; ACEi, Angiotensin-converting enzyme inhibitor; ARB, angiotensin-receptor blockers.

## Data Availability

The datasets used and/or analyzed during the current study are available from the corresponding author on reasonable request.

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
