# Peer review of "Diagnostic Performance of Circulating miRNAs and Extracellular Vesicles in Acute Ischemic Stroke"

_ijms, 2022, doi:10.3390/ijms23094530_

Round 1

Reviewer 1 Report

In this study authors have looked at expression of miRNA mir-19-3p,-186-5p, let7f and EV in IS patients. The authors using pcrs and other analysis and, have concluded that miR-19a-3p is appropriate biomarker for IS.

The manuscript is well written and study is well designed. However few things need to be addressed before the paper is accepted.

1) Please mention the number of replicates and patients samples used for PCRs in all the figure legend and method section. 

2) Figure legends are missing many key details. Please fix it

3) In Figure-1 authors has shown platelet activity in response to ADP, RAP and AA, is platelet activity in control group of control test(white bars) and control group of AA-induced significantly different?  Please change control test to unstimulated(I believe white bar represents unstimulated samples)

4)In Fig-2 authors have shown change in miR expression upon activation. What was the internal control used to normalized the sample. What samples did authors used for miRNA extraction? Platelet or Plasma? 

5) In fig3) Authors have shown platelet activation and EV concentration, and there is significant change in expression of CD62, CD45. How do authors made sure that there were no contaminating membrane debris in samples and these changes are not due to change in cellular debris. 

Author Response

Dear Reviewer,

Best Regards,

Reviewer 2 Report

The authors are presenting an innovative and clinically relevant manuscript on an important topic. Inflammatory response and platelet related reactions after acute ischemic stroke have been studied but to my knowledge, microRNAs response has never been analyzed.

The manuscript is structured, in this respect, following previously published data to choose relevant Mirns. I see some similitude with other works on microparticles (extracellular vesicles) after diving and also after DCS see: Thom SR, Bennett M, Banham ND, Chin W, Blake DF, Rosen A, Pollock NW, Madden D, Barak OF, Marroni A, Balestra C, Germonpre P, Pieri M, Cialoni D, Le PJ, Logue C, Lambert D, Hardy KR, Sward D, Yang M, Bhopale VM & Dujic Z. (2015). Association of Microparticles and Neutrophil Activation with Decompression Sickness. J Appl Physiol (1985) 119, 427-434.

I do believe that some parts of the discussion will be of interest to refine the discussion, a number of studies have been performed by Stephen Thom team, they worked pretty well on the same direction as yours (although not in stroke) and some of the pathological cascades and responses are parallel to yours.

In the introduction about MMP-9, I remember a paper showing that MMP-9 could be increased transiently by a short oxidative stress (Cimino et al 2012).

I actually find the manuscript very interesting and agree with the fact that it is innovating, not many researchers have focused on that particular issue. Some microRNAs in relation with inflammation and transient oxidative stress have just been recently investigated (Bosco et al 2021) without finding the same Mirns as yours, I would be interested, if you think that it is useful for the paper, it is a suggestion, some sentences in the discussion on this.

Thank you for the opportunity of reading this manuscript.

Author Response

Dear Reviewer,

Best regards.

Reviewer 3 Report

The manuscript by Eyileten et al. describes an approach based on circulating molecules and particles for the diagnosis of IS. The approach appears confused, as it is unclear how miRs were selected and it is unclear the benefit of a diagnosis relying onto different and complex methodologies (qRT-PCR and flow cytometry), which makes the approach time-consuming and demanding in terms of instrumentation foa pathology in need of rapid diagnosis. In addition, the manuscript does not provide evidence that the proposed biomarkers can discriminate different types of strokes, that were not enrolled in the study.

Major points

In the Abstract, the sentence “In the current study we aimed to determine miRNAs (miR-19a-3p, miR-186-5p and let-7f) that we previously selected by means of a bioinformatic approach, as well as we evaluated the concentrations of EVs from platelets, leukocytes, and endothelial cells to analyse their diagnostic and predictive utility in patients with acute IS.” is unclear. What do authors mean for bioinformatic approach? In the Abstract, but also on the Introduction, it is not specified what type of approach was carried out, despite the reference to a previous study. An in silico selection strategy has to be introduced, not simply defined a “bioinformatic approach”. Where were the miRNAs evaluated? It is not possible to understand (circulating miRNA? EVs associated?)

In the Results section of the Abstract, the response to AA is reported, but there is description of this type of experiment in methods. The sentence should be eliminated, also because the Abstract is too long and do not focus on main findings

In the Abstract, it is stated that platelet-EVs  are CD62+, in the Discussion it is stated that platelet-EVs  are CD61+. This generates confusion. Besides platelet-EVs (CD62+) and leukocyte-EVs (CD45+) concentrations appears quantitatively dissimilar, but as platelet-EVs represent most of circulating EVs, it is unclear why the number of leukocyte-EVs CD45+ is higher (Table 1). Also platelet-EVs (CD61+) concentration is similar to leukocyte-EVs CD45+, but one would expect that it should be higher. Authors should comment on the reliability of these markers to select for platelet and leukocytes specific EVs, as they are possibly not specific.

Minor points

In the Abstract, “Increased inflammation activated blood coagulation system…” should be “Increased inflammation activates blood coagulation system…”

The sentence “MiRNAs are stored 83 in platelets, and they can regulate platelet reactivity by directly and indirectly targeting 84 specific genes and therefore, modifying protein synthesis” is misleading, as it seems to suggest that miRNAs are typically present in platelets and not in other cell types. The sentence “EVs were found to be associated with chronic inflammatory 90 diseases as well as acute inflammatory events such as acute IS.” is also misleading, as EVs have been shown to be released by any kind of cell and have been shown to be associated with a variety of conditions, not only with inflammation.

Author Response

Dear Reviewer,

Best regards.

Round 2

Reviewer 3 Report

The manuscript by Eyileten et al. is now better organized, but still lacks accuracy

Major points

In the Background, it is stated that authors “aimed to assess circulating miRNAs profile”. However, authors did not profile miRNA, but selected a panel of miRNA and assessed their level. The sentence should more accurate, otherwise it suggests a miRNA sequencing effort, but this was not the case. The same applies to the ends of Methods section “was used to determine circulating plasma miRNAs”, as not all circulating miRNA were determined.

In the Limitations section, authors now specify that “Moreover, regarding EVs evaluation, due to low positive events count (<10, which is too little counts for a reliable concentration estimation) CD62+ and CD61, CD62+ events were omitted from the analysis in the control group, and they were presented only in the IS patients’ groups.”. So, it seems that there is a problem in detecting CD62+ platelet-derived EVs in controls, even if platelet-EVs are well represented in circulating EVs. This point is confusing and should be clarified. Is CD62 exclusively expressed in platelets or other factors could affect this result? As for example, CD62+ behaviour is more like that of CD45+ than to that of CD61+. These findings should be discussed in a clearer and deeper manner.

Author Response

Dear Reviewer,

We are thankful for the time and effort spent by you to provide such an in-depth review and critique of our manuscript. We have corrected the manuscript to comply with the Reviewers’ comments. Parts of the manuscript that were added or changed during the review process are highlighted in yellow to help you identify the changes. In this response, we have fully addressed all the points of the Reviewers comments. The corrections are listed below.

  1. In the Background, it is stated that authors “aimed to assess circulating miRNAs profile”. However, authors did not profile miRNA, but selected a panel of miRNA and assessed their level. The sentence should more accurate, otherwise it suggests a miRNA sequencing effort, but this was not the case. The same applies to the ends of Methods section “was used to determine circulating plasma miRNAs”, as not all circulating miRNA were determined.

Answer: We are sorry for our careless mistake, we have corrected them

  1. In the Limitations section, authors now specify that “Moreover, regarding EVs evaluation, due to low positive events count (<10, which is too little counts for a reliable concentration estimation) CD62+ and CD61, CD62+ events were omitted from the analysis in the control group, and they were presented only in the IS patients’ groups.”. So, it seems that there is a problem in detecting CD62+ platelet-derived EVs in controls, even if platelet-EVs are well represented in circulating EVs. This point is confusing and should be clarified. Is CD62 exclusively expressed in platelets or other factors could affect this result? As for example, CD62+ behaviour is more like that of CD45+ than to that of CD61+. These findings should be discussed in a clearer and deeper manner.

Answer: We thank the Reviewer for asking this question.

Regarding EVs detection we collaborated with Dr. Edwin Van Der Pol (University of Amsterdam, Vesicle Observation Center), who is a coauthor of methodological guidelines of EVs studies with a long-term expertise. And we believe that there was no EVs detection problem that occurred during the flow cytometry analysis.

[1] https://www.ahajournals.org/doi/10.1161/CIRCRESAHA.117.309417?url_ver=Z39.88-2003&rfr_id=ori:rid:crossref.org&rfr_dat=cr_pub%20%200pubmed

[2] https://pubmed.ncbi.nlm.nih.gov/?sort=date&term=van%20der%20Pol%20E&cauthor_id=26563735

As the Reviewer pointed out this issue, we clarified it in the limitations section, as follows: 

‘Moreover, regarding EVs evaluation, due to low positive events count (<10, which is too little counts for a reliable concentration estimation) CD62+ and CD61,CD62+ events were omitted from the analysis in the control group and they were presented only in the IS patients groups. It is important to note that, CD62+ is a marker for platelet activation, hardly present in plasma of healthy individuals, but abundantly present in plasma of patients with myocardial infarction [75]. Therefore, hardly detected CD62+ platelet-derived EVs in the control group can be due to normal platelet function. Thus, we still suggest that CD62+-exposing EVs are a better marker differentiating patients with IS and controls than CD61+ or CD45+, because the latter are present also in plasma of healthy individuals, albeit at lower concentrations than in plasma of IS patients’.

Round 3

Reviewer 3 Report

The manuscript by Eyileten et al. is now improved, as suggested corrections have been made